# Developmental Delays in Socio-Emotional Brain Functions in Persons with an Intellectual Disability: Impact on Treatment and Support

**DOI:** 10.3390/ijerph192013109

**Published:** 2022-10-12

**Authors:** Tanja Sappok, Angela Hassiotis, Marco Bertelli, Isabel Dziobek, Paula Sterkenburg

**Affiliations:** 1Berlin Center for Mental Health in Developmental Disabilities, Ev. Krankenhaus Königin Elisabeth Herzberge, 10365 Berlin, Germany; 2Division of Psychiatry, University College London, London W1T 7BN, UK; 3Camden and Islington NHS Foundation Trust, London NW1 0PE, UK; 4CREA (Research and Clinical Centre), San Sebastiano Foundation, Misericordia di Firenze, 50142 Florence, Italy; 5Clinical Psychology of Social Interaction, Humboldt-Universität zu Berlin, 10099 Berlin, Germany; 6Bartiméus, 3941 XM Doorn, The Netherlands; 7Department of Clinical Child and Family Studies, Vrije Universiteit Amsterdam, 1081 BT Amsterdam, The Netherlands

**Keywords:** developmental neuroscience, emotional functioning, intellectual disability, intervention mental health, limbic system, social brain network

## Abstract

Intellectual disability is a neurodevelopmental disorder with a related co-occurrence of mental health issues and challenging behaviors. In addition to purely cognitive functions, socio-emotional competencies may also be affected. In this paper, the lens of developmental social neuroscience is used to better understand the origins of mental disorders and challenging behaviors in people with an intellectual disability. The current concept of intelligence is broadened by socio-emotional brain functions. The emergence of these socio-emotional brain functions is linked to the formation of the respective neuronal networks located within the different parts of the limbic system. Thus, high order networks build on circuits that process more basic information. The socio-emotional skills can be assessed and complement the results of a standardized IQ-test. Disturbances of the brain cytoarchitecture and function that occur at a certain developmental period may increase the susceptibility to certain mental disorders. Insights into the current mental and socio-emotional functioning of a person may support clinicians in the calibration of treatment and support. Acknowledging the trajectories of the socio-emotional brain development may result in a more comprehensive understanding of behaviors and mental health in people with developmental delays and thus underpin supports for promotion of good mental health in this highly vulnerable population.

## 1. Introduction

According to a nationwide US survey, one in six persons has a developmental disability, with most suffering from neurodevelopmental disorders, such as attention-deficit/hyperactivity disorder (ADHD) (9.5%), autism spectrum disorders (2.5%), learning disability (7.9%), or intellectual disability (1.2%) [1]. During the past decade, these prevalence rates have increased, e.g., from 1.1% to 2.5% for autism and from 0.9% to 1.2% for intellectual disability [1,2] as a result of better assessment and diagnosis. According to the WHO Global Health metrics, there are about 100 million people with an intellectual disability world-wide [3].

People with an intellectual disability are more vulnerable to physical or mental disorders [4,5,6,7]. According to the recent meta-analysis of Mazza [4], 33,6% suffer from a mental disorder, which is about double of the prevalence rates in the general population. In 2018, one in three persons with a cognitive disability experienced more than 50% physically unhealthy days, and one in two persons reported more than 50% mentally unhealthy days [7]. Furthermore, approximately one in two persons with a cognitive disability slept less than 6 h a night, had experienced depression, and rated their health as fair to poor [7]. Especially these chronic and secondary health conditions decrease the life expectancy of persons with an intellectual disability by around 20 years compared to the general population [8,9]. Factors associated with death in people with an intellectual disability are mental and physical illnesses (e.g., cancer, heard, pulmonary and renal diseases) and physical disabilities such as cerebral palsy and epilepsy [10]. More than a third of those deaths are potentially amenable to healthcare interventions [11,12]. Due to the poorer management of chronic health conditions in primary healthcare, application of reasonable adjustments in services and awareness training have to be expanded to meet the needs of people with developmental disabilities [12]. 

People with an intellectual disability face various barriers to receiving adequate healthcare. Hence, person-related factors, such as lack of information and health literacy or communication difficulties, and system-related aspects, such as insufficient knowledge or even discriminatory attitudes among practitioners and service providers, have to be considered [10]. Health impairments lead to a significant emotional, social, and financial burden on the patients and on their social networks. In the European Union, intellectual disability was amongst the top ten most expensive diseases of the brain [13], requiring an annual expenditure of 43.3€ billions in 2010. In a Canadian study, adults with an intellectual disability were nearly four times more likely to incur high annual healthcare costs than those without an intellectual disability [14]. An Australian study demonstrated the considerable economic impact of intellectual disability on families, governments, and broader society [15], and a study on health-service use and the costs of Americans with an intellectual disability revealed that the presence of chronic medical conditions and poor mental health status predicted high expenses across various types of healthcare [16]. Taken together, all the evidence points to the urgent need for public health to focus on effective interventions for and the holistic management of secondary disorders in people with an intellectual disability. 

The present paper develops five main themes. First, the current understanding of intelligence and the options of broadening the concept by integrating socio-emotional competencies will be outlined. Second, the development of socio-emotional brain functions will be linked to the maturation of the socio-emotional brain networks. Third, options to systematically assess socio-emotional brain functions will be offered. Fourth, the relatedness of developmental neuroscience and mental health in persons with an intellectual disability will be addressed. Finally, the implications of the developmental approach to treatment and support will be posed. 

The lens of developmental neuroscience provides insights into the co-regulating processes of emotion regulation, social interaction and adaptation and may serve as a window into general psychological mechanisms. In a translational pathway, the underlying processes may have a broad impact on our understanding of human behavior [17].

## 2. Intellectual Disability Revisited: About the Current Understanding of Intelligence and Why There Is a Need for Broadening the Concept towards the Social Realm

In Western societies, Descartes [18] laid the foundation for continental rationalism: “Je pense, donc je suis” [‘I think therefore I am’]. Pioneers such as Damasio [19] overcame the dualistic body-mind distinction and delineated the importance of emotion and the sensorial body within cognitive science approaches. Socio-emotional and cognitive processes have been proposed to be highly intertwined and overtly present in all physiological and pathological mental activities. Ciompi [20] and the RDoC (Research Domain Criteria) framework [21] provided well-known examples. In the International Classification of Diseases, ICD-11, disorders of the intellectual development (also termed “intellectual disability” in DSM-5) are counted among the neurodevelopmental disorders that “arise during the developmental period” and “involve significant difficulties in the acquisition and execution of specific intellectual, motor, language, or social functions” [22]. As such, in addition to poor cognitive functions, social, affective, and adaptive functions are also affected. However, the concept of “intelligence” is still dominated by Descartes’ rationalism, and the diagnosis of an intellectual disability is still centred on general IQ measures, academic learning, and complex logical-deductive executive functions, such as working memory, processing speed, attention, encoding, verbal comprehension and expression, abstract reasoning, and problem solving. 

However, emotional and social intelligence differ from cognitive intelligence and require different neural circuitries [23]. Socio-emotional brain functions refer to the ability to regulate one’s own emotional expression, to identify the emotional expressions of others, to interpret emotional cues, to respond accordingly, and to self-soothe and manage emotional outbursts. Emotional intelligence is therefore essential for the quality of relationships with the self and with other people, the ability to adapt, to cope with stress and to communicate and interact appropriately [24]. A functional account of emotions posits that emotions are mental activities that respond to environmental stimuli, such as a social or physical challenge, and determine in turn physical and behavioral reactions. Emotional brain functions also influence cognitive processes, including perception, attention, learning, memory, reasoning, and problem solving and vice-versa. A detailed knowledge about emotional competencies such as emotional awareness, managing of own emotions and emotions of other people, self-motivation, impulse control and empathy may support caregivers, clinicians, and therapists to get an insight into the mental processes and needs of persons with low cognitive abilities [23,24]. 

In this theoretical paper, we aim to integrate knowledge from developmental neuroscience to expand our understanding of intellectual disability. As intellectual disability is a disorder of the brain that manifests during the developmental period, the stepwise acquisition of the different emotional and social skills can be depicted alongside the developmental trajectories of typically developing children [25,26], c.f. Table 1. 

## 3. Linking the Socio-Emotional Brain Functions with the Maturation of the Socio-Emotional Brain Networks 

Depending on the functionality of the respective brain networks, different ways of thinking and developmental tasks, needs, and skills can be learned [25,26]. Knowledge of the socio-emotional brain network functions may provide the clinician with insights into the “inner world” of people who experience difficulties expressing their own thoughts/feelings, and it could increase the understanding of their behaviors [25,26,27]. We therefore wish to outline the developmental milestones of brain development and link it to the respective socio-emotional functions. Persons with a developmental delay principally follow the same trajectories as people with a neurotypical development; however, the developmental milestones may be reached later or incompletely.

The brain architecture is scaffolded prenatally and early in life, followed by an extended period of differentiation of the cytoarchitecture by dendritic growth and the formation, pruning, and stabilization of synapses. While short-range connectivity predominates in infancy, there is a shift towards long-range networks in adolescents and adults [25]. The developmental changes of structural brain connectivity result from a sequence of genetic and epigenetic mechanisms at key developmental stages [25,28,29,30]. Environmental factors and early life experiences in social interactions play a crucial role in the coordination and timing of the specific neuronal patterning [25,31,32]. 

At birth, the neuronal networks are already at a certain stage of maturity. However, functional magnetic resonance imaging studies showed that local brain activity and functional connectivity differ between neonates and adults. While in the neonatal brain, for example, sensorimotor, visual and auditory areas were most active, other frontal brain regions, the basal ganglia and limbic/paralimbic areas showed lower dynamic connections than adult brains [28]. The emergence of the various socio-emotional skills is linked to the formation of the respective brain networks located within the different parts of the limbic system [33]. Thus, higher order networks build on circuits that process more basic information [25]. The recognition of emotional-communicative signals, for example, is a prerequisite for the proper functioning of the Theory of Mind network, or stress and emotion regulation abilities are necessary for successful impulse control. 

Figure 1 exemplifies the stepwise development of the Theory of Mind network.

The different steps (Figure 1) provide an insight into the stepwise emerging mentalization abilities. Depending on the respective developmental stage, different ways of thinking occur [32]: The first step is the action-oriented way of thinking also called the ‘teleological thinking’. At this stage, feelings and thoughts cannot be expressed with words but goal-oriented actions. Self-injurious or destructive behaviors can represent a common response to frustration.The second step is the ‘concrete thinking’ stage. In this stage, persons cannot discriminate between their own thoughts and the thoughts of other persons, meaning that thinking is reality to them: “The way I think, the way it is”.The third step is thinking in the pretend mode of thinking ‘pseudomentalization’. In this stage, the inner world (fantasy/imagination) is disconnected from the outside world. The interpretation of a situation is unrelated to the reality of other people. People in the pretend mode of thinking may have problems with feeling emotions in the ‘here and now’ and use clichés and empty words.In a further step, the individual is able to acknowledge that other persons have different feelings, thoughts, intentions, and motivations: The Theory of Mind network is developed and the person is able to mentalize (c.f. Figure 1).

Knowledge of the development of the mentalization abilities is supportive for a proper understanding of a person’s mental framework, especially in the case of the non-mentalizing ways of thinking. 

Depending on a variety of factors, including specific brain alterations and their time course, in persons with an intellectual disability, the formation of the socio-emotional brain networks may differ from the pattern that can be observed in typical development [35,36]. According to the definition in the ICD-11/DSM-5, intellectual disability begins during the developmental period and is associated with impairments of the different brain functions, which will be related to the different neural networks in the following stage.

However, when the developmental delay is severe, the *deep limbic system* drives most emotional and relational acts. This part of the limbic system develops prenatally and during the very first months after birth and includes the central nucleus of the amygdala, the hypothalamus, and parts of the brain stem including the periventricular grey and the vegetative nuclei. This first step of brain development is accompanied by an action-oriented *way* of thinking with an inability to express feelings and thoughts with words [32]. Feelings and thoughts cannot yet be expressed with words but involve goal-oriented actions. The autonomic and the stress-regulation systems process basal functions for survival, such as heart rate and temperature control, feeding, sexuality, territoriality, and stress responses including fight-flight reactions (see Figure 2) [32]. These mostly unconscious processes are genetically-epigenetically determined and influenced by early life experiences [37].

In moderate to severe forms of developmental delay, the functions located within the *mesolimbic system* determine the way of thinking and social interaction. This part of the limbic system is located in the basolateral amygdala, the ventral tegmental area, and the nucleus accumbens/ventral striatum and is the seat of the reward and the reward expectation system where emotional conditioning and emotion regulation are processed. Hence, basic emotional functions, such as fear, sadness, disgust, happiness, and anger, are determined [31,38]. The basic needs are safety and security [25,26]. In this stage of brain development, the person is learning to build up an inner picture of the outside environment (object permanence) and to experience his/her own thoughts as reality (concrete thinking) [39]. This may result in misunderstandings, as facts are not differentiated from convictions [32]. In the next step, accompanied by the ability to differentiate between the self and the other, the pretend mode of thinking ‘pseudo-mentalization’ arises [32]. This can lead to meaningless conversations, such as repetitive questioning, and in the case of trauma, may result in dissociation [32]. The mesolimbic system develops within the first months and years of life and operates predominantly unconsciously (cf. Figure 2) [33]. 

In milder forms of developmental delays, the *upper limbic system* dominates social cognition and adaptive behaviors. It is composed of a group of tightly interconnected cortical brain areas including the prefrontal, the orbitofrontal, the ventromedial frontal, the anterior cingulate, and the insular cortex. It comprises the neural activity that controls the Theory of Mind, different aspects of executive functions, risk assessment, and reality awareness [40,41,42]. In a top-down mechanism, these neocortical networks attenuate the emotional responses of the lower-order brain circuits located within the mesolimbic and deep limbic systems [43]. Logical thinking, impulse control, delayed gratification, and affect regulation are important for pro-social behaviors [44]. Concomitantly, emotional states, such as empathy, friendship, loyalty, and moral thinking, may be observed [27,45,46]. The upper limbic system evolves in late childhood and adolescence and can be partly modulated by learning [33].

A variety of conditions associated with an intellectual disability, such as ASD, meningoencephalitis, or genetic syndromes, may cause impairments of the early wiring within the limbic system and the associated brain functions [25,42,45,47,48]. Therefore, the socio-emotional brain functions, e.g., perspective-taking skills, may differ in persons with different syndromes (Cornelia de Lange syndrome vs. William syndrome) or comorbidities such as ASD or attention-deficit-hyperactivity disorders (ADHD) [34,49]. Furthermore, stress and trauma can influence social brain functions in general, and specifically for persons with intellectual disabilities [25,26,50].

So far, there is a lack of assessment instruments for emotional and social competencies. Thus, incorporating structured information about the social, emotional, and practical skills into the assignment to the different levels of intellectual disability may be supportive for clinical care, especially when it comes to the more severe forms where classical IQ-tests cannot be applied. 

## 4. Assessment of Socio-Emotional Functioning

Depending on the individual pattern of developmental delay, also in adults with an intellectual disability, socio-emotional brain functions may be delayed. For a comprehensive evaluation of the mental abilities of a person, we propose integrating the assessment of the different emotional and social skills located in the respective brain network, as these are crucial for perception, the way of thinking, and adaptive behaviors outcomes. Currently, IQ tests focus on logical-deductive academic skills [23,24,51]. However, the IQ score does not always relate to the individual’s functioning at a specific point in time, and emotional competences, such as affect regulation, risk assessment, delayed gratification, impulse control, mentalizing abilities, and reality awareness, must also be taken into account [24,46]. Structured assessments addressing these abilities may be helpful to further ascertain this population and support clinicians in the calibration of treatment and support.

Being aware that development is a continuous process, for an assessment of the functional skills, a stepwise model is necessary. Researchers have developed assessment instruments to determine the socio-emotional functioning of a person with an intellectual disability [52,53]. In particular, the Scale of Emotional Development-Short (SED-S) is based on the normative developmental trajectory of the social brain network to define the central characteristic of socio-emotional functioning in a certain age group [35]. The instrument was tested for proof of evidence for criterion validity on item, domain, and scale level by applying the scale to a sample of typically developing children [54]. For the majority of items, the expected response pattern emerged, showing the highest response probabilities in the respective target age groups. Agreement between the classification of the different SED-S domains and the chronological age of children with normative development was high (*κw* = 0.95; exact agreement = 80.6%) [54]. Interrater reliability at domain level ranged from *κw* = 0.98 to 1.00, and internal consistency was high (α = 0.99) [54]. The SED-S is applicable and valid in children [55] and adults with ID [34]. Lower levels of socio-emotional development are associated with more severe forms of challenging behaviors [49]. Mentalization abilities can be assessed using the Reflective Functioning Questionnaire–Mild to Borderline intellectual Disabilities *+* (RFQ-MBID) [56]. Depending on the pattern of brain alterations, in persons with an intellectual disability, the intellectual reference age is likely to be distinct from the emotional age [49]. Therefore, we argue that the socio-emotional brain functions should be evaluated separately, and specific instruments should be added to those already in use to measure the IQ itself. The utility of the comprehensive assessment of the level of socio-emotional brain functions should be viewed as paramount in supporting clinicians in personalizing treatment and care in the clinical setting. A person’s social, emotional, and practical abilities are central for the adaptive behavior, emotional well-being, and mental health.

## 5. Mental Disorders and Developmental Neuroscience in Persons with an Intellectual Disability

Mental health and emotional functioning in a particular social environment are strongly interrelated. Persons with an intellectual disability are highly vulnerable to mental disorders [4,5,6]. The conceptualization of psychopathology and mental disorders relies on the grouping of defined symptoms into syndromes that yield a psychiatric diagnosis. The co-occurrence of certain developmental disorders and intellectual disability may suggest a common underlying neurobiology at an early stage of brain development. The increasing evidence for shared genetic etiology across different psychiatric disorders and intellectual disability suggests a continuum of neurodevelopmental causality that includes both the heterogeneity and the overlap of risk factors and disease mechanisms [57]. The developmental miswiring within the social brain networks at sensitive periods may be associated with mental disorders that occur at a certain point of brain development [45,58]. Hence, certain disorders such as autism spectrum disorders (ASD) may be more prevalent in people with more severe forms of intellectual disability while other disorders such as social anxiety disorders may be more often seen in milder forms of intellectual disability.

In ASD, for example, core symptoms, such as perspective-taking skills, are rooted in developmental delays of the brain circuits related to social cognition [59]. In ADHD, widespread alterations of structural and functional brain connectivity are described [60]. Insecure attachment appears to be linked to social experiences that occur during critical periods of development which affect the architecture of the limbic and stress regulation system and have an impact on emotion processing, emotion regulation, and risk assessment [23,25,61]. Alterations in specific neural circuits that develop prenatally or very early in life are also often reported in persons with an intellectual disability, especially when the developmental delay is severe [45,58]. 

However, other psychiatric disorders, such as social anxiety disorders, dissociative disorders, or personality disorders, may require the maturation of higher-order social brain networks and so cannot be found earlier than age 5 years [62,63,64]. Social anxiety disorders require perspective-taking skills located in the Theory of Mind network [62]. Dissociative disorders are associated with subcortical white matter alterations within the higher limbic system [63,64]. Conduct disorders may progress to antisocial personality disorders during adolescence/early adulthood [65]. Social anxiety, dissociative disorders, or personality disorder can be linked to disturbances of higher-order brain circuits and typically arise concomitantly to the formation of the respective neural networks during childhood and adolescence. These mental disorders are rarely observed in severe forms of intellectual disability and are more prevalent in persons with borderline intellectual functioning or mild cognitive impairments [5]. 

Therefore, it can be argued that disturbances of the brain cytoarchitecture and function that occur at a certain developmental period may increase the susceptibility to certain mental disorders. This is supported by research examining emotional intelligence and psychopathy [66,67]. The developmental approach for socio-emotional brain functions in persons with an intellectual disability offers a fundamental perspective in mental health and opens up new treatment options [45]. 

## 6. Impact of the Social Brain Development on Treatment and Support

The quantity and quality of studies evaluating the efficacy of psychological therapies in persons with an intellectual disability and mental ill-health are still limited, especially in those with severe to profound intellectual disabilities [68]. Some studies tested commonly used psychosocial interventions, such as cognitive behavioral therapy, and there are valuable efforts to adapt the methods to the level of cognitive functioning; however, treatment manuals for severe and profound levels are still scarce [49,69]. In addition, effectiveness studies often exclude persons with multiple disabilities and comorbidities which is the clinical reality we are faced with [49,68]. 

With regard to treatment and care, aspects such as the level of socio-emotional functioning and the associated mental competencies and possibilities for reflection may support the decision for or against a certain therapeutic approach. Well-developed perspective-taking skills, for example, may increase the probability of the individual deriving benefit from cognitive behavioral therapy or mentalization-based treatment, while individuals with limited stress regulation abilities may be more likely to respond to bodily and experience-based treatment methods, such as attachment-based behavioral therapy or dance and movement therapy [48,70,71,72]. Targeting evidence-based treatment programs that are personalized and in line with the individual’s abilities and goals is particularly vital in persons with developmental disabilities [49,73]. The ‘social information processing model’, for example, aims to choose the type of intervention according to the mental state of a person during a social interaction [74]. Moreover, knowledge of the emotional reference age of the individual may enable caregivers to be more attuned to his/her emotional needs and, therefore, promote and maintain good mental health. Finally, the awareness of the socio-emotional functioning of persons with an intellectual disability at that time point may enhance the diagnostic process of ascertaining co-occurrent psychiatric disorders. Externalizing behaviors or observable psychological distress may be interpreted as psychopathological symptoms but could be better explained as a mismatch between the level of individual development expected for the chronological age and the level of actual individual functioning [56,75]. This is particularly useful in persons with low or absent verbal communication skills, in whom key elements of psychiatric disorders, such as delusions, hallucinations, or suicidal ideation, are often very hard to recognize and may only be expressed by changes in behavior [76]. Matson et al. [77] claimed that “accurately identifying the causes of adaptive skill deficits will likely result in more precise and effective treatment” (p. 1317). Therefore, disturbances of the socio-emotional brain networks at a certain developmental period may increase the susceptibility to certain mental disorders. Aligning treatment options according to the level of socio-emotional functioning may strengthen the efficacy and increase the outcome of treatment of certain mental disorders. Furthermore, teaching emotional competencies may further improve skills such as emotional awareness, managing of own emotions and emotions of other people, self-motivation and empathy [78].

## 7. Discussion

The developmental perspective on the socio-emotional brain network may give insights into their own perspective and experiences, especially in people who experience difficulties expressing their own thoughts and feelings, and it may support clinicians to better understand the shown behaviors [25,26,27]. Therefore, the linkage of the developmental milestones of brain development with the respective socio-emotional functions may help adapt treatment and support accordingly. We are aware that the staged limbic-structure theory is simplifying the complexity of human brain development [79,80]. This perspective or framework opens up the road for rolling out and promoting early intervention strategies that impact both behavior and adaptive skills as these appear to be likely modifiable factors that can improve longer term outcomes. Further in-depth insights into the perceptual, cognitive, and social-communicative functions in specific syndromes like Downs syndrome or Williams syndrome need to be considered [81,82]. 

## 8. Conclusions

The focus of this article is to connect recent knowledge from developmental neuroscience with clinical research in persons with an intellectual disability with the aim of deducing the implications for treatment and support. Despite the given limitations of the broad-brush description of the stepwise development of the brain, specifically of the different parts of the limbic system and its associated functions, a developmental neurobiological basis may offer an additional perspective in our understanding and conceptualization of psychopathology and mental health in persons with developmental delays. The developmental miswiring within the socio-emotional brain networks at sensitive periods may be associated with mental disorders that occur at a certain point of brain development [45,58]. A common underlying neurobiology at an early stage of brain development may cause an association of certain disorders with different severities of intellectual disability. This synthesis offers relevant evidence about the necessity to integrate developmental neuroscience into clinical practice and care for persons with an intellectual disability to further promote mental health in this highly vulnerable population. Knowledge of development of the socio-emotional brain is important in the clinical and daily work context, as it provides insights into the inner world of persons who may have difficulties in reporting about their own thoughts and needs. Accordingly, this article aims to cross the bridge from basic neuroscience to the practical work with persons with developmental disabilities. A comprehensive assessment of intellectual functioning including socio-emotional functioning is important in treatment provision that is personalized and addresses individual goals and deficits. Therapeutic considerations should not only contribute to increased well-being but should also be consistent with the person’s emotional status and congruent with the social environment (cf. Figure 1). This extended understanding of how people with intellectual disability function and how they participate in society may enable persons with developmental disabilities to “participate in every aspect of life to the best of their abilities and desires” [7]. We assert that it is only in this way that the person can be supported to fully realize his/her potential and prevent new onset or exacerbations of a mental disorder. 

## Figures and Tables

**Figure 1 ijerph-19-13109-f001:**
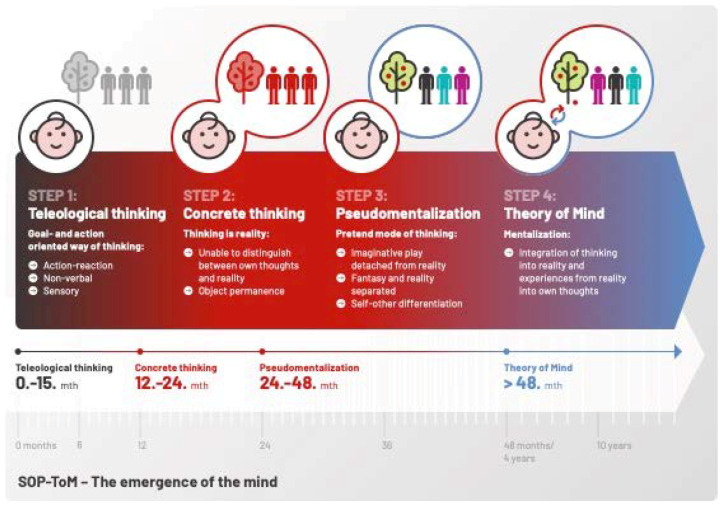
Milestones of development of the Theory of Mind network: Teleological thinking (0–15 months); concrete thinking (15–24 months); Pseudomentalization (24–48 months); Theory of Mind network (>48 months). Details c.f. [32,34].

**Figure 2 ijerph-19-13109-f002:**
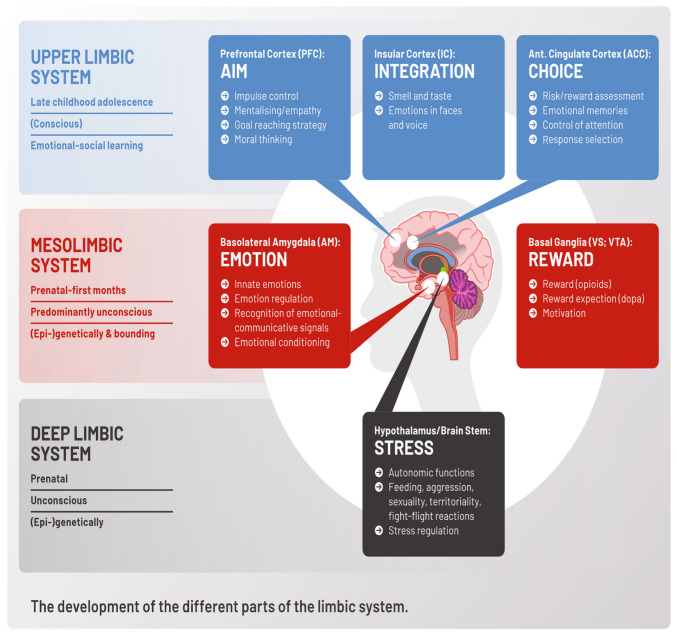
The development of the different parts of socio-emotional brain networks. Details about the different parts and functions of the limbic system c.f. [33]. Socio-emotional brain functions rely on a proper formation of the respective brain areas of the limbic system. Depending on the severity of brain damage, also socio-emotional competencies are affected accordingly. Abbreviations: PFC: Prefrontal Cortex; IC: Insular Cortex; Ant.: Anterior; ACC: Anterior Cingulate Cortex; Bl.: Basolateral; AM: Amygdala; Basal Gang.: Basal Ganglia; VS: Ventral Striatum; VTA: Ventral Tegmental Area.

**Table 1 ijerph-19-13109-t001:** Development of socio-emotional brain functions [26].

Level of Socio-Emotional Development (Reference Age)	Corresponding Level of Intellectual Functioning	Socio-Emotional Developmental Milestones
Adaptation (0–6 months of age)	Profound intellectual disability (F73)	Integration of sensory information and external stimuli (place, time and people), processing of stimuli, regulation of physical processes
3. Socialisation 4. (7–18 months of age)	Profound intellectual disability (F73)	Social bonds, object permanence, rough body scheme
5. First Individuation (19–36 months of age)	Severe-profound intellectual disability (F72–F73)	Self-Other differentiation, recognizing and expressing one’s own will.
6. Identification 7. (4–7 years of age)	Moderate-severe intellectual disability (F71–F72)	Ego formation, change of perspective (Theory of Mind), interaction with peers, differentiation between fantasy and reality
8. Reality awareness (8–12 years of age)	Mild-moderate intellectual disability (F70–F71)	Moral action, assessment of one’s own abilities, self-differentiation, awareness of reality, logical thinking
9. Social Individuation (13–18 years of age)	Borderline-mild intellectual disability, typical intelligence	Abstract thinking skills, sexual identity, self-reflection, independence, responsibility, identity formation, moral self

## Data Availability

Not applicable.

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
