# Peer review of "Developmental Delays in Socio-Emotional Brain Functions in Persons with an Intellectual Disability: Impact on Treatment and Support"

_ijerph, 2022, doi:10.3390/ijerph192013109_

Round 1
Reviewer 1 Report
Dear Editor,
The manuscript entitled," Developmental Delays in Socio-Emotional Brain functions in Persons with an Intellectual Disability: Impact on Treatment and Support" reviews socio-emotional brain development and highlights the related disabilities with brain networks. This review supports the importance of considering intellectually disabled individuals' social integration by understanding the biological basis of disability. However, there are some questions which needs to be answered.
1. For a review article IMRAD categorization in abstract is not relevant (e.g., methods, results, …).
2. In section 2 (Current understanding of intelligence), the differences between emotional intelligence and IQ have not been discriminated clearly. In following, the contribution of these two types of intelligence has not been explained in individuals with low cognitive abilities. This issue builds a gap in defining Socio-emotional brain functions in the next paragraph. The are some articles which can be helpful for delineation of these two concepts (DOIs: 10.1093/brain/awg177; 10.1016/j.jad.2017.04.012; https://doi.org/10.1037/a0031746; https://doi.org/10.1057/s41599-022-01173-w; 10.3928/00220124-20110921-03).
3. Figure 1 needs redesigning. As, in the upper row (Brain networks) only classified limbic area which mainly involves emotional networks. However, mental and cognitive functioning (Middle row) is associated with cortical networks. Then, depicting related neocortical networks such as temporal association cortex, prefrontal cortex, orbitofrontal cortex, and other cortical areas in developing mental and intellectual functioning are recommended (https://doi.org/10.1016/j.dcn.2020.100850). Also, the resolution of figure 1 is low and it has two repeated captions, and it's better to use colored version.
4. the resolution of figure 2 is low and it's better to use colored version.
5. Misspelling, In line 55, pulmonary? "and diseases", which type of diseases?
6. The language consistency needs to be followed in writing. For example, both of the behavior or behaviour was used in the manuscript.
Author Response
Review 1
The manuscript entitled, "Developmental Delays in Socio-Emotional Brain functions in Persons with an Intellectual Disability: Impact on Treatment and Support" reviews socio-emotional brain development and highlights the related disabilities with brain networks. This review supports the importance of considering intellectually disabled individuals' social integration by understanding the biological basis of disability. However, there are some questions which needs to be answered.
Answer:
We are grateful for the overall comment and the suggestions made by the reviewer. We think they highly improved the quality of the article.
- For a review article IMRAD categorization in abstract is not relevant (e.g., methods, results, …).
Answer:
The categories were deleted in the abstract:
“Abstract: Intellectual disability is a neurodevelopmental disorder with a relevant co-occurrence of mental health issues and challenging behaviors. In addition to pure cognitive functions, also socio-emotional competencies may be affected. In this paper, the lens of developmental social neuroscience is used to better understand the origins of mental disorders and challenging behaviors in people with an intellectual disability. The current concept of intelligence is broadened by socio-emotional brain functions. The emergence of these socio-emotional brain functions is linked to the formation of the respective neuronal networks located within the different parts of the limbic system. Thus, high order networks build on circuits that process more basic information. The socio-emotional skills can be assessed and complement the results of a standardized IQ-test. Disturbances of the brain cytoarchitecture and function that occur at a certain developmental period may increase the susceptibility to certain mental disorders. Insights into the current mental and socio-emotional functioning of a person may support clinicians in the calibration of treatment and support. Acknowledging the trajectories of the socio-emotional brain development may results in a more comprehensive understanding of behaviors and mental health in people with developmental delays and thus underpin supports for promotion of good mental health in this highly vulnerable population.”
- In section 2 (Current understanding of intelligence), the differences between emotional intelligence and IQ have not been discriminated clearly. In following, the contribution of these two types of intelligence has not been explained in individuals with low cognitive abilities. This issue builds a gap in defining Socio-emotional brain functions in the next paragraph. The are some articles which can be helpful for delineation of these two concepts (DOIs: 10.1093/brain/awg177; 10.1016/j.jad.2017.04.012; https://doi.org/10.1037/a0031746; https://doi.org/10.1057/s41599-022-01173-w; 10.3928/00220124-20110921-03).
Answer:
We highly appreciate the supportive suggestions and implemented the literature as follows:
- Bar‐On, R.; Tranel, D.; Denburg, N.L.; Bechara, A. Exploring the neurological substrate of emotional and social intelligence,Brain, 2003, 126(8), 1790–1800,https://doi.org/10.1093/brain/awg177 and
- Pisanos, D.E. Emotional Intelligence: It’s more than IQ. The Journal of Continuing Education in Nursing 2011, 21, DOI: 10.3928/00220124-20110921-03:
“However, emotional and social intelligence differ from cognitive intelligence and require different neural circuitries [23]. Socio-emotional brain functions refer to the ability to regulate one’s own emotional expression, to identify the emotional expressions of others, to interpret emotional cues, to respond accordingly, and to self-soothe and manage emotional outbursts. Emotional intelligence is therefore essential for the quality of relationships with the self and with other people, the ability to adapt, to cope with stress and to communicate and interact appropriately [35]. A functional account of emotions posits that emotions are mental activities that respond to environmental stimuli, such as a social or physical challenge, and determine in turn physical and behavioral reactions. Emotional brain functions also influence cognitive processes, including perception, attention, learning, memory, reasoning, and problem solving and vice-versa. A detailed knowledge about emotional competencies such as emotional awareness, managing of own emotions and emotions of other people, self-motivation, impulse control and empathy may support caregivers, clinicians, and therapists to get an insight into the mental processes and needs of persons with low cognitive abilities [23; 35].”
- Copestake, S., Gray, N. S., & Snowden, R. J. Emotional intelligence and psychopathy: A comparison of trait and ability measures. Emotion, 2013, 13(4), 691–702. DOI: 10.1037/a0031746:
“Therefore, it can be argued that disturbances of the brain cytoarchitecture and function that occur at a certain developmental period may increase the susceptibility to certain mental disorders. This is supported by research examining emotional intelligence and psychopathy [67].”
- Tuyakova, U., Baizhumanova, B., Mustapaeva, T. et al. Developing emotional intelligence in student teachers in universities. Humanit Soc Sci Commun 9, 155 (2022). https://doi.org/10.1057/s41599-022-01173-w:
“Furthermore, teaching emotional competencies may further improve skills such as emotional awareness, managing of own emotions and emotions of other people, self-motivation and empathy [78].”
- Figure 1 needs redesigning. As, in the upper row (Brain networks) only classified limbic area which mainly involves emotional networks. However, mental and cognitive functioning (Middle row) is associated with cortical networks. Then, depicting related neocortical networks such as temporal association cortex, prefrontal cortex, orbitofrontal cortex, and other cortical areas in developing mental and intellectual functioning are recommended (https://doi.org/10.1016/j.dcn.2020.100850). Also, the resolution of figure 1 is low and it has two repeated captions, and it's better to use colored version.
Answer:
As suggested by the reviewer, we redesigned Figure 1 and focused on the stepwise development of the Theory of Mind network. We agree that the former information summarized in Figure 1 was overloaded with information and mixed different concepts. Moreover, we used a colored version and a high-resolution file:
Due to the focus on the Theory of Mind network we added a paragraph which comprehensively explains the different milestones of the developmental trajectory of mentalization:
“The different steps (Figure 1) provide an insight into the stepwise emerging mentalization abilities. Depending on the respective developmental stage, different ways of thinking occur:
- The first step is the action-oriented way of thinking also called the ‘teleological thinking’. At this stage, feelings and thoughts cannot be expressed with words but goal-oriented actions. Self-injurious or destructive behaviors can represent a common response to frustration.
- The second step is the ‘concrete thinking’ stage. In this stage, persons cannot discriminate between their own thoughts and the thoughts of other persons meaning that thinking is reality to them: “The way I think, the way it is”.
- The third step is thinking in the pretend mode of thinking ‘pseudomentalization’. In this stage, the inner world (fantasy/imagination) is disconnected from the outside. The interpretation of a situation is unrelated to the reality of other people. People in the pretend mode of thinking may have problems with feeling emotions in the ‘here and now’ and use clichés and empty words.
- In a further step, the individual is able to acknowledge that other persons have different feelings, thoughts, intentions, and motivations: The Theory of Mind network developed and the person is able to mentalize (c.f. Figure 1).
Knowledge of the development of the mentalization abilities is supportive for a proper understanding of a person’s mental framework, especially in case of the non-mentalizing ways of thinking.”
We thank the reviewer also for the literature suggestion. We added it in the text as follows:
“At birth, the neuronal networks are already at a certain stage of maturity. However, functional magnetic resonance imaging studies showed that local brain activity and functional connectivity differ between neonates and adults. While in the neonatal brain, for example, sensorimotor, visual and auditory areas were most active, other frontal brain regions, the basal ganglia and limbic/paralimbic areas showed lower dynamic connections than adult brains [26].”
- the resolution of figure 2 is low and it's better to use colored version.
Answer:
The resolution was low due to problems in uploading the figure. We try to improve the uploading process and use a colored version of Figure 1:
- Misspelling, In line 55, pulmonary? "and diseases", which type of diseases?
Answer:
Thanks for this hint. We corrected the misspelling and added the missing word:
“Factors associated with death in people with an intellectual disability are mental and physical illnesses (e.g. cancer, heard, pulmonary and renal diseases) and physical disabilities such as cerebral palsy and epilepsy [10].”
- The language consistency needs to be followed in writing. For example, both of the behavior or behaviour was used in the manuscript.
Answer:
We checked the language throughout the manuscript and adapted according to US spelling consistently.

Reviewer 2 Report
The article is innovative in that it presents a broad overview of the knowledge of developmental morphology and related epigenetic and social factors in the development of not only pathological but also typological characteristics of an individual. Even if some of the presented conclusions are speculative, they can be a stimulus for a more detailed analysis in a number of subsequent works.
Author Response
We highly appreciate the overall comment made by the reviewer and agree with his statement. We indeed hope to stimulate scientists and practitioners working the field of intellectual disabilities to further improve the quality of support and mental health of the affected people.

Round 2
Reviewer 1 Report
The revised manuscript (# ijerph-1933755) was improved from the aspect of discussing the concept of intelligence broadly. Also, other issues considered by the authors carefully.